# Optimisation of Glycerol and Itaconic Anhydride Polycondensation

**DOI:** 10.3390/molecules27144627

**Published:** 2022-07-20

**Authors:** Krzysztof Kolankowski, Magdalena Miętus, Paweł Ruśkowski, Agnieszka Gadomska-Gajadhur

**Affiliations:** Faculty of Chemistry, Warsaw University of Technology, Noakowskiego 3 Street, 00-664 Warsaw, Poland; krzysztof.kolankowski.dokt@pw.edu.pl (K.K.); magdalena.mietus.stud@pw.edu.pl (M.M.); pawel.ruskowski@pw.edu.pl (P.R.)

**Keywords:** unsaturated glycerol polyesters, poly(glycerol itaconate), design of experiments

## Abstract

Glycerol polyesters have recently become objects of interest in tissue engineering. Barely known so far is poly(glycerol itaconate) (PGItc), a biocompatible, biodegradable polyester. Due to the presence of a C=C electron-acceptor moiety, it is possible to post-modify the product by Michael additions to change the properties of PGItc. Thus, using PGItc as one of the elements of cellular scaffold crosslinked in situ for bone tissue regeneration seems to be a very attractive yet unexplored solution. This work aims to optimize the synthesis of PGItc to obtain derivatives with a double bond in the side chain with the highest conversion rates. The experiments were performed with itaconic anhydride and glycerol using mathematical planning of experiments according to the Box-Behnken plan without solvent and catalyst. The input variables of the process were the ratio of the OH/COOH, temperature, and reaction time. The optimised output variables were: the degree of esterification (ED_titr_), the degree of esterification calculated from the analysis of ^1^H NMR spectra (ED_NMR_), and the degree of itaconic anhydride conversion—calculation based on ^13^C NMR spectra (%X_13C_^NMR^). In each of statistical models, the significance of the changed synthesis parameters was determined. Optimal conditions are when OH/COOH ratio is equal to 1.5, temperature is 140 °C and time of reaction is 5 h. The higher OH/COOH ratio, temperature and longer the experiment time, the higher the value of the degree of esterification and the degree of anhydride conversion.

## 1. Introduction

Glycerol polyesters are a new group of polymers with great potential in medicine. They are biocompatible and biodegradable [1]. Depending on the chain length in dicarboxylic acid, they are characterised by various degradation time in the human body [2]. These polymers can be used short-term (days) or long-term (years) [1,3,4]. Among the polyesters with high biomedical potential can be distinguished, for example poly(glycerol sebacate), poly(glycerol succinate), poly(glycerol fumarate), poly(glycerol maleate), and poly(glycerol itaconate) [5,6,7,8,9,10].

The substrate for producing these polymers is glycerol, a non-toxic and non-irritating trihydroxy alcohol [11]. Due to the presence of three hydroxyl groups in the structure of glycerol, this compound is hygroscopic and highly soluble in water [12]. Glycerol is the molecular skeleton of fats in the human body. It has antibacterial activity, at which the maximum is observed at 36 °C (ca human body temperature) [11,12]. It is used in many industries such as food (as a sugar substitute for sweetening beverages), pharmaceutical (drug delivery systems) and cosmetics (as a body care substance) [11,13,14].

The best known and most widely reported in literature is a glycerol-based polyester, poly(glycerol sebacate)—PGS. PGS is a synthetic and biodegradable polymer obtained by the polycondensation of sebacic acid with glycerol (Figure 1) [5,15,16,17,18,19,20,21,22]. PGS exhibits not only biocompatibility but as well resorbability [23]. Due to its unique properties, PGS is being studied by many researchers for future use in tissue regeneration, both soft (e.g., heart) and hard (e.g., bone) or in the encapsulation of anti-cancer drugs [15,16,19,22,24,25].

The PGS structure consists of hydrophobic (eight methylene groups) and hydrophilic (hydroxyl groups from glycerol) groups [24]. It allows the creation of structures that bind drugs of both hydrophobic and hydrophilic characters [19,24].

A more hydrophilic material is poly(glycerol succinate) (PGSu) [26]. This polymer can be synthesised not only in the reaction between succinic acid and glycerol but also succinic anhydride and glycerol (Figure 2) [26,27,28,29,30]. Like PGS, the synthesis reaction of PGSu occurs without the use of solvent or catalyst [26,31,32,33]. The functional properties of PGSu are similar to those of PGS. Succinic acid is a natural metabolite in the Krebs cycle; therefore, it is non-toxic to mammalian organisms [34]. The FDA (Food and Drug Administration)-approved succinic acid as a component of medical and pharmaceutical devices [35].

An attractive alternative to using saturated dicarboxylic acids in synthesising polyesters is the use of dicarboxylic acids containing unsaturated bonds (Figure 3) [36,37,38]. The use of maleic acid, fumaric acid or itaconic acid may be an example of a substrate for the preparation of innovative polymers [8].

The double bond is an important element in the poly(glycerol maleate) (PGMal) structure, whose presence enables post-polymerisation reactions [39]. Poly(glycerol fumarate) (PGF) is an isomer of poly(glycerol maleate) (Figure 4) [40]. As in PGMal, a C=C double bond moiety is present in its structure. Poly(glycerol fumarate) can be obtained by reacting glycerol with fumaric acid or fumaric anhydride. Another method to obtain PGF is the reaction of glycerol with maleic anhydride. The Z-mers isomerise to E-mers with increasing temperature and time of the process [41].

Poly(glycerol itaconate) (PGItc) is a curious material because the double bond occurs not in the main chain but in the side chain. It can significantly affect the properties and reactivity of the product compared to PGMal [42,43]. Over the past 30 years, there has been a significant interest in itaconic acid (Figure 5).

Itaconic acid can be produced by fermentation by filamentous fungi (most commonly *Aspergillus terreus*) [42,43]. The production volume of acid itself is not significant (due to competitive reactions producing other acids such as maleic or fumaric acid). Still, due to emerging new applications of its derivatives, an increase in demand for itaconic acid is assumed. In 2025, itaconic acid production could be as high as 170 kton/year [42,43].

Itaconic acid is used in medical applications as a hardening agent for contact lenses and as dental cement [44]. Due to its anti-inflammatory and antimicrobial properties, it is believed that itaconic acid can be used to produce various types of drugs [45,46]. In 2020, Wang and his researchers conducted an experiment designed to test the effects of itaconic acid on proteins present in living cells and performing important functions. Itaconic acid caused the modification of a large number of proteins. There were changes in regulatory pathways responsible for the body’s immune response and changes in protein structures responsible for cell death. The polymers based on itaconic acid have a potential for biomedicine applications, e.g., cellular scaffolds. The reaction of itaconic acid or anhydride with glycerol may result in the formation of poly(glycerol itaconate) (PGItc) (Figure 6) [47,48].

There is a lack of publications directly focused on poly(glycerol itaconate). In contrast, there are many articles describing research on polyitaconates, e.g., based on alkyl groups. Such a compound is, e.g., poly(dodecyl itaconate), which was the subject of a 2015 study led by S. Ramakrishnan [49]. The polymer was obtained by a two-step process. The first step of reaction is conducted with a catalyst (dibutyltin dilaurate, DBTDL) at 150 °C under a nitrogen atmosphere. The oligomerisation reaction involving dibutyl itaconate and 1,12-dodecanediol has taken place. The reaction was then conducted under reduced pressure at 160 °C in the presence of quinol, which was used to prevent the formation of an insoluble polymer. After isolating the product, pure poly(dodecyl itaconate) was obtained.

Itaconate polyesters form rubber-like polymers upon crosslinking. They do not exhibit very high strength, but they can be used in tissue engineering as drug delivery hydrogels [50]. The structure of the growing polymer can be controlled relatively simply by selecting the molar ratio of glycerol to acid or anhydride. The temperature and time of polycondensation influence the degree of esterification and branching, molecular weight, viscosity, and mechanical properties of PGItc.

## 2. Results and Discussion

### 2.1. FTIR and NMR Analysis

The structure of the obtained polyester was confirmed by the FTIR spectrum (Figure 7). 

The PGItc spectrum shows a broad band of 3500–3100 cm^−1^ characteristic for hydroxyl group vibrations. The bands 2953 and 2892 cm^−1^ correspond to vibrations of C-H bonds in the main aliphatic chain. The band evidences the presence of an unsaturated C=C bond at 1638 cm^−1^. The bands at 1709, 1176, and 1036 cm^−1^ are, respectively, the vibration bands of the carbonyl group, acyl, and alkoxy groups. 

The interpretation of the ^1^H NMR spectra (Figure 8) enabled the confirmation of the product structure and relevant calculations.

Glycerol protons are present in the 5.50–4.40 ppm and 4.30–3.20 ppm ranges, and their chemical shift depends on how glycerol is substituted (possible linear, terminal, and dendritic esters). Signals in the range 4.40–4.20 ppm are protons of unreacted glycerol. From 2.3 ppm to 1.9 ppm are CH_2_ group protons. The effect of Ordelt saturation occurring (which means the attack of glycerol OH groups on the double bonds) is observed in the range of 3.10–2.53 ppm. The conversion rate of this reaction was calculated and is in the range of 14.2–34.5% (Appendix A). We also observe the isomerisation of itaconic fragments to mesaconic fragments. This process occurs with an efficiency of about 1.2–9.8% (Appendix A). The higher the process temperature is, the more considerable side reactions become.

Our particular attention was drawn to the signals in the unsaturated bond area. We assigned a defined origin to each of them (Figure 8). The spectra registered for pure reactants were especially useful. 

The following formula was used to calculate the esterification degree using NMR spectra.
ED_NMR_ = ((∫P_itc_ + ∫M_itc_)/(∫An_itc_ + ∫M_itc_ + ∫P_itc_)) × 100%(1)
where 

∫P_itc_—The value of the integral of the signal is from the itaconic polyesters, oligoesters, monoesters;

∫M_itc_—The value of the integral of the signal is from the itaconic monoesters;

∫An_itc_—The value of the integral of the signal is from the itaconic anhydride.

On the ^13^C spectrum (Figure 9), the signals of carbonyl carbons (173–164 ppm), the signals of double bond carbons (136–127 ppm), and the signals of glycerol moiety carbons (76–60 ppm) are observed in sequence.

The following formula was used to calculate the itaconic anhydride conversion degree,
%X_13C_^NMR^ = ((∫A + ∫B + ∫C + ∫D)/(∫A + ∫B + ∫C + ∫D + ∫An_itc_)) × 100%(2)
where

∫A + ∫B + ∫C + ∫D—The value of the integrals of the signals is from the itaconic polyesters, oligoesters, and monoesters; 

∫An_itc_—The value of the integral of the signals is from the itaconic anhydride.

### 2.2. Statistical Analysis

The Box–Behnken plan was used to create mathematical models, as it is the most popular and convenient method of describing the process. The matrix plan consists of 15 experiments, 3 of which are performed under identical conditions to check the repeatability of conditions and the experimenter’s skill.

Such mathematical modelling provides a great deal of valuable information about the object under study and reduces the number of experiments from 27 to 15 (for 3 input variables), which is incredibly precious when scaling up. This saves time and money, which are crucial from the viewpoint of factory economics. The ratio of functional groups, temperature, and time were chosen as input variables because, in our opinion, these variables are easy to control and have the greatest impact on the price of the synthesised product.

The experimentally calculated and model-calculated values of the output variables are summarised in Table 1.

Based on the Pareto chart analysis (Appendix A), we determined which coefficients of the regression equation are significant. We concluded that only the product of the input variables *x*_1_ and *x*_3_ (t_calculated_ = 3.27) is significant (t_calculated_ > t_critical_). It was found that among the input variables, the linear relationship *x*_1_*x*_3_ has the greatest effect on *y*_1_, so the value of the variable x_2_ was set as a constant equal 1).

The equation that describes the degree of esterification defined by titrations methods (*y*_1_) is:*y*_1_ = 49.2 + 14.4 × *x*_1_ × *x*_3_(3)

The response surface is the graphic presentation of the calculated model (Figure 10).

Based on the results of the F-test, the adequacy of the model used was determined. The value of F for the equation with one significant variable was 14.94 (F_calculated_ < F_critical_), so the applied model can be considered adequate (Appendix A).

The coefficient R^2^ is 0.53. It means that although only the *x*_1_*x*_3_ relationship is significant; the other input variables and their relationships, although not significant, affect the output variable *y*_1_. 

The esterification values obtained from the titrations were compared with those obtained from the model (Table 1). The experimentally obtained values differ from the approximated values by ±15.1 percentage points, although for some experiments, the differences are minor (±0.5 percentage points).

Obtainment of the highest esterification degree is possible when the process is conducted for 5 h with the 1.5 OH/COOH ratio or 3 h and 0.5 OH/COOH ratio— ED_tit_ > 60.0%.

Running the process using a ratio of 1.5 (excess glycerol hydroxyl groups) is associated with a higher ED_tit_, meaning that linear rather than branched products are more likely to be formed. Running the process for 5 h at a functional group ratio of 0.5 results in an ED_tit_ < 35.0%.

Pareto chart analysis (Appendix A) contributed to the conclusion that the input variable *x*_1_ (t_calculated_ = 7.56) has a significant effect on the output variable *y*_2_ (t_calculated_> t_critical_). The variable *x*_3_ has the least significant effect on the output variable *y*_2_. Therefore, the time value was set as a constant, *x*_3_ = 1. Although the input variable *x*_2_ was insignificant, it was included in the regression equation because the effect score was just below the critical value for the significance level of *p* = 0.05. The addition of the variable *x*_2_ to the regression equation contributed to an increase in the R^2^ coefficient from 0.72 to 0.81.

The equation that describes the degree of esterification determined from the analysis of ^1^H NMR spectra (*y*_2_) takes the following form:*y*_2_ = 68.2 + 2.21 × *x*_1_ − 0.738 × *x*_2_(4)

Using the regression equation, the response surface graph was plotted (Figure 11).

Based on the F-test, we concluded that the calculated value of F_calculated_ for the equation with significant variables is 4.96 and <F_critical_. Thus, the used model is adequate (Appendix A).

The ED_NMR_ values obtained from the experiments were compared with the values obtained calculated by the model (Table 1). They differ by only ±1.4 percentage points. The coefficient of determination R^2^ is 0.81.

A high ED_NMR_ value (>69.5%) can be obtained by running the reaction at 140 °C with an OH/COOH ratio equal to 1.5. By conducting the reaction at 110 °C with a functional group ratio of 0.5, the ED_NMR_ value is less than 67.5%. The ED_NMR_ < 65.5% was achieved by conducting the reaction at a functional group ratio of 0.5 at 140 °C.

The Pareto chart analysis determined the significance of the regression equation coefficients (Appendix A). Only the variable *x*_1_ (t_critcal_ = 3.41) (t_calculated_ > t_critical_) has a statistically significant effect on the degree of itaconic anhydride conversion. The input variable *x*_2_ was also included in the regression equation. However, it was taken into the regression equation due to the substantial increase in the coefficient of R^2^ from 0.29 to 0.44.

The equation is of the form:*y*_3_ = 57.1 + 5.35 × *x*_1_ −3.76 × *x*_2_(5)

This equation was used to plot the dependence of the degree of itaconic anhydride conversion (*y*_3_) on the temperature (*x*_2_) and the ratio of functional groups (*x*_1_) for *x*_3_ = 1 (Figure 12).

The F_calculated_ value for the equation with significant variables is 6.32 < F_critical_, so the model used is adequate (Appendix A).

The variability of the output variable y_3_ is also influenced by other variables and their relationship, although they were insignificant.

The %X_13C_^NMR^ values obtained from the model differ from the approximated values by ±13.0 percentage points, but for some experiments, the differences are ±0.1 percentage points (Table 1). 

The process should be run at the highest functional group ratio at the lowest temperature to obtain the highest %X_13C_^NMR^ value. A high %X_13C_^NMR^ value cannot be obtained despite high-temperature usage when the functional group ratio is 0.5.

### 2.3. Experiment under Optimal Conditions

Poly(glycerol itaconate), characterised by the highest possible values of the output variables, was obtained by the response utility profile function software (available in Statistica). The utility of the values of output variables was determined (low, medium, and high utility values—Appendix A). The highest values obtained in the experimental plan were used as the low utility values of the output variables. We determined that the highest degrees of conversion could be obtained if the synthesis was conducted at a functional group ratio of 1.5 (*x*_1_ = 1) at 140 °C (*x*_2_ = 1) for 5 h (*x*_3_ = 1) (Appendix A).

The experiment was carried out under the assumed conditions, summarising the analysis results in Table 2.

The calculated value of the degree of ED_tit_ esterification is larger than the result calculated from the profile of approximated values by more than ten percentage points. Despite the difference, it can be considered that this value is not significantly different from the expected value. The calculated experimental value of ED_NMR_ differs by 0.7 percentage points from the value calculated using the profile. The experimentally calculated value of the degree of the itaconic anhydride conversion (%X_13C_^NMR^) differs from the value calculated using the profile by only 0.2 percentage points. 

Received values indicate a good fit of the statistical models to reality.

## 3. Materials and Methods

### 3.1. NMR

A nuclear magnetic resonance (NMR) spectroscopy was used. A total of 130–150 mg of product was dissolved in 1 mL of DMSO d-6 (Deutero GmbH, Kastellaun, Germany). The mixture was shaken for 24 h and then transferred to an NMR tube. NMR spectra were obtained using an Agilent 400 MHz spectrometer. ^13^C NMR spectra were collected without the nuclear Overhauser effect.

### 3.2. FTIR

IR spectra were obtained in ATR technics using ALPHA II BRUKER spectrometer. For each sample, 32 scans in the range 400–4000 cm^−1^ were performed and averaged.

### 3.3. Acid Number

A total of 0.5–1.5 g of the sample was weighed and dissolved in 25.00 mL of methanol. Then, it was titrated with 1M aqueous NaOH solution until the indicator (thymol blue) turned from yellow to blue. The acid number (AN) was calculated using the following formula:AN [mg_KOH_/g_sample_] = ((V − V_0_) × M_NaOH_ × 56.1)/m(6)
where

V, the volume of 1 M NaOH solution used to titrate the sample;

V_0_, the volume of 1 M NaOH used for blank titration;

M_NaOH_, the titer of the solution for the titration (1 M);

56.1, the molar mass of KOH;

m, sample weight.

The final result is the average of three determinations.

### 3.4. Ester Number

A total of 0.2–0.5 of the sample was weighed and dissolved in 15.00 mL of methanol and 20 mL of 1 M aqueous NaOH solution. The prepared solutions were refluxed for 1 h. Then, the mixture was cooled to room temperature. The excess NaOH was titrated with a 1 M aqueous solution of hydrochloric acid against phenolphthalein until it became discoloured. The ester number (EN) was calculated using the following formula:EN [mg_KOH_/g_sample_] = (((V_0_ − V) × M_HCl_ × 56.1)/m) − AN(7)
where 

V, the volume of 1 M HCl solution used to titrate the sample;

V_0_, the volume of 1 M HCl used for blank titration;

MHCl, the titer of the solution for the titration (1 M);

56.1, the molar mass of KOH;

m, sample weight.

The final result is the average of three determinations. 

### 3.5. Esterification Degree

The ED was calculated according to the following formula:ED = EN/(EN + AN) × 100%(8)
where: EN, ester number; AN, acid number.

### 3.6. Statistical Analysis

Calculations and graphics were made in Statistica 13.1 (StatSoft, Cracow, Poland).

### 3.7. Synthesis Procedure

PGItc syntheses were carried out in a Mettler Toledo MultiMax parallel reactors system in Hastelloy reactors. Glycerol (≥99%, Sigma Aldrich, Burlington, MA, USA), and itaconic anhydride (99%, Ambeed, Arlington Heights, IL, USA) were used without prior preparation (Figure 13). Glycerol (10.62 g, 0.115 mol; 13.53 g, 0.147 mol; 6.45 g 0.070 mol) and itaconic anhydride (19.38 g, 0.173 mol; 16.47 g 0.147 mol; 23.55 g 0.210 mol) were weighed into the reactor in amounts depending on the molar ratio of the functional groups. Each time, the sum of the reactants was 30.00 g.

Reactors were equipped with a mechanical stirrer, temperature sensor, and Dean-Stark apparatus. In the first stage, the mixture was heated for over 20 min to temperature *x*_2_. The temperature was held constant for *x*_3_ hours. After the reaction, the mixture was cooled down to room temperature.

### 3.8. Optimisation Process

Experiments were planned according to the mathematical methods of planning experiments according to the Box–Behnken plan. The natural values of the variables were coded as −1, 0, or +1 according to Table 3.

## 4. Conclusions

Three mathematical models were developed to consider the influence of the changed parameters (ratio of functional groups, temperature, and time) on the investigated values (degree of esterification determined by titration methods, degree of esterification received based on ^1^H NMR spectra, and degree of itaconic anhydride conversion received based on ^13^C NMR spectra). All of the obtained models were adequate. The relationship between the ratio of functional groups and the running time of the process has a significant effect on the ED_titr_ value. To obtain the highest ED_titr_ value, the reaction should be carried out for the shortest time (3 h) using the lowest ratio of functional groups (0.5) or carried out for 5 h at a ratio of functional groups of 1.5. Only the ratio of the functional groups of substrates used significantly affects the ED_NMR_. However, a value close to significant is demonstrated by the temperature variable. The reaction should be carried out at the lowest possible temperature (110 °C) with the highest functional group ratio (1.5) to obtain a high ED_NMR_ value. This conclusion seems illogical and may occur due to the low variability of the ED_NMR_ value. The %X_13C_^NMR^ value is significantly affected only by the functional group ratio. Since the significance level is a conventional value, the effect of temperature was also considered in determining the regression equation. The regression equation has demonstrated that to obtain a high itaconic anhydride value, the reaction should be carried out at the highest possible ratio of functional groups (1.5), and the process temperature should be 110 °C. 

The experiment was carried out under optimal conditions, i.e., functional group ratio 1.5, temperature 140 °C, and time 5 h, in which ED_titr_ = 62.3%, ED_NMR_ = 70.4%, and %X_13C_^NMR^ = 62.8% were obtained. The values of the obtained variables compared to the predicted values show a good fit of the statistical model to reality. 

It could be interesting to extend the experimental area to increase the initial variables’ variability and obtain polyester with higher molecular weight. The following research stage will be viscosity and cellular studies of the obtained materials.

## Figures and Tables

**Figure 1 molecules-27-04627-f001:**
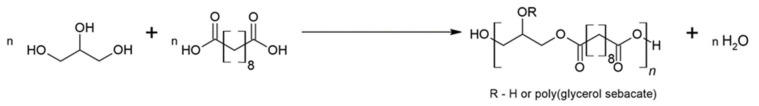
Polycondensation of sebacic acid with glycerol.

**Figure 2 molecules-27-04627-f002:**
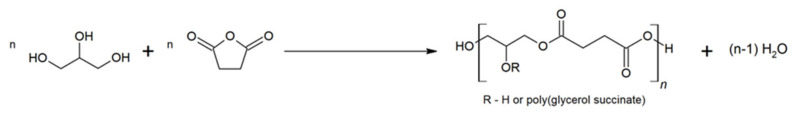
Synthesis of PGSu from succinic anhydride and glycerol.

**Figure 3 molecules-27-04627-f003:**
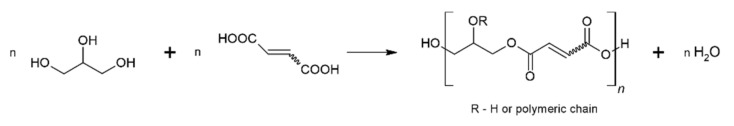
Schematic representation of the synthesis of polyesters with unsaturated bonds.

**Figure 4 molecules-27-04627-f004:**
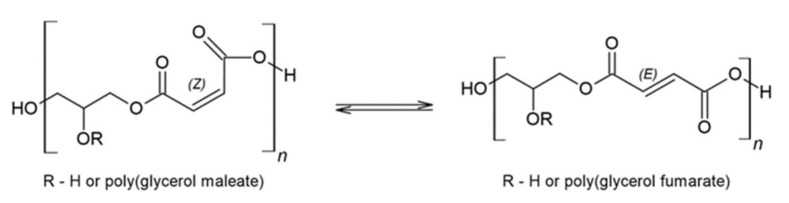
Isomerisation of PGMal and PGF.

**Figure 5 molecules-27-04627-f005:**
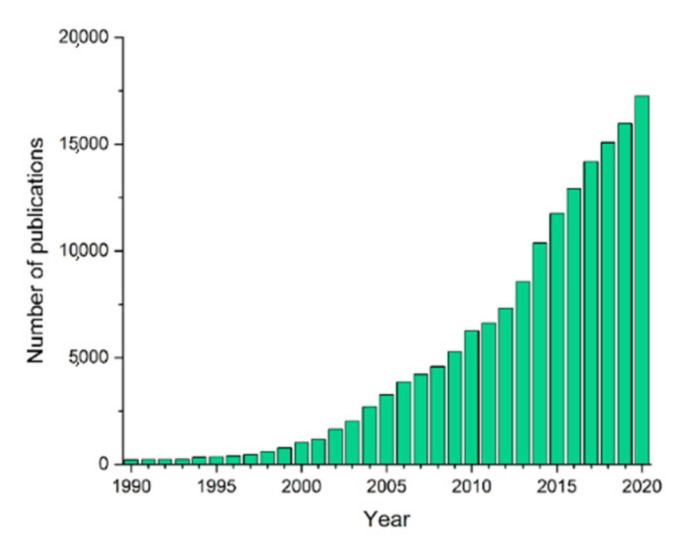
Number of publications with the keyword ”itaconic acid” in the years 1990–2020—own elaboration based on data from PubMed.

**Figure 6 molecules-27-04627-f006:**
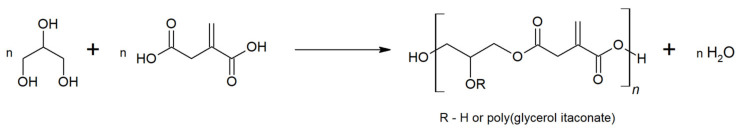
Synthesis of PGItc from glycerol and itaconic acid.

**Figure 7 molecules-27-04627-f007:**
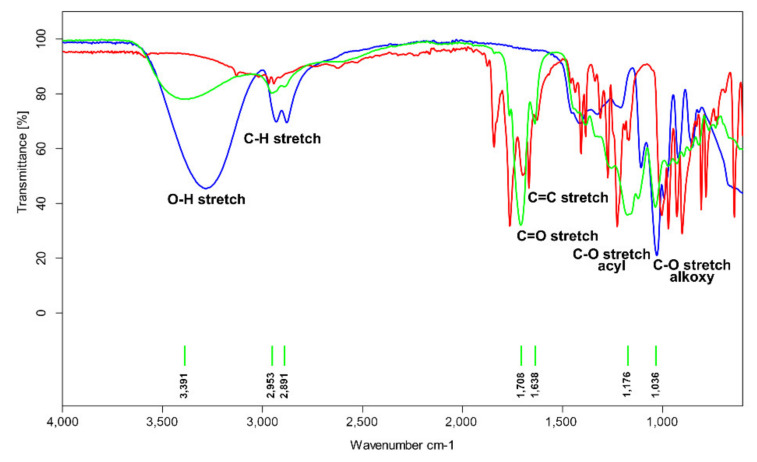
FTIR spectra of poly(glycerol itaconate) (green), itaconic anhydride (red), and glycerol (blue).

**Figure 8 molecules-27-04627-f008:**
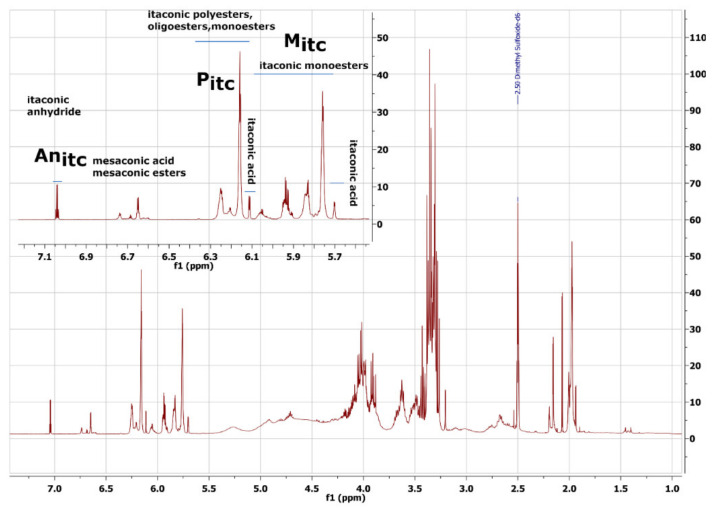
The ^1^H NMR spectra of poly(glycerol itaconate).

**Figure 9 molecules-27-04627-f009:**
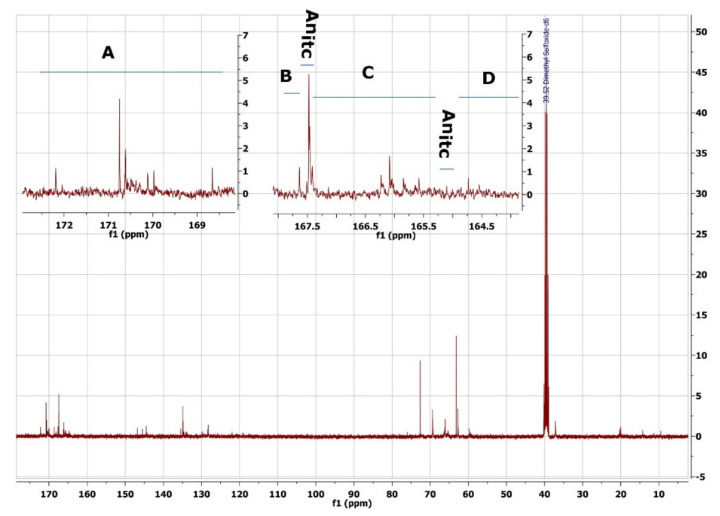
^13^C NMR spectrum of poly(glycerol itaconate).

**Figure 10 molecules-27-04627-f010:**
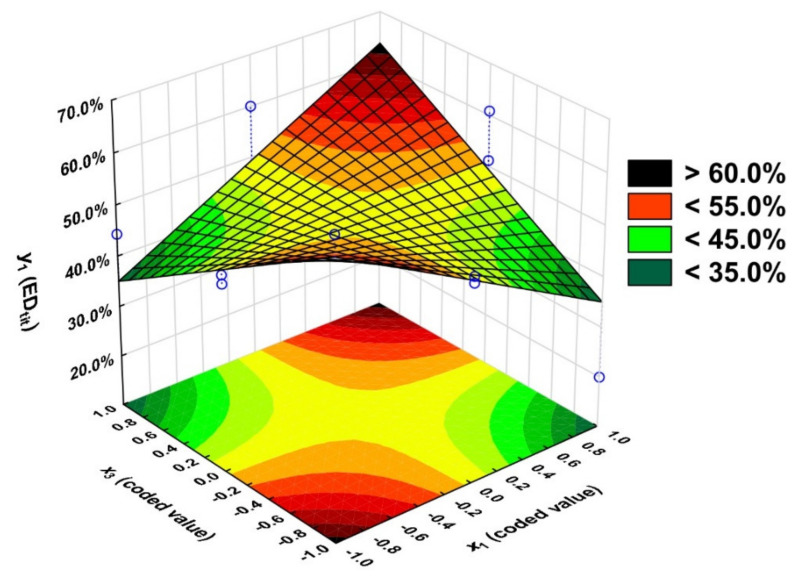
Dependence of esterification degree (ED_tit_) of PGItc, on the OH/COOH ratio (*x*_1_) and the time (*x*_3_), *x*_2_ = 1.

**Figure 11 molecules-27-04627-f011:**
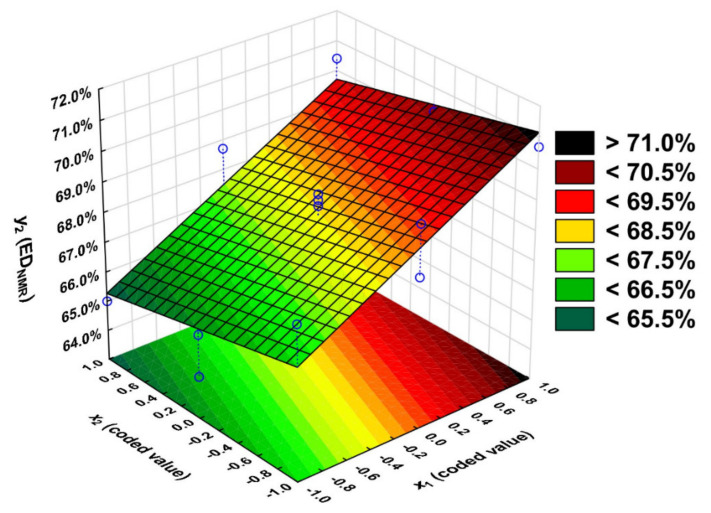
Dependence of esterification degree (ED_NMR_) of PGItc, on the OH/COOH ratio (*x*_1_) and the temperature (*x*_2_), *x*_3_ = 1.

**Figure 12 molecules-27-04627-f012:**
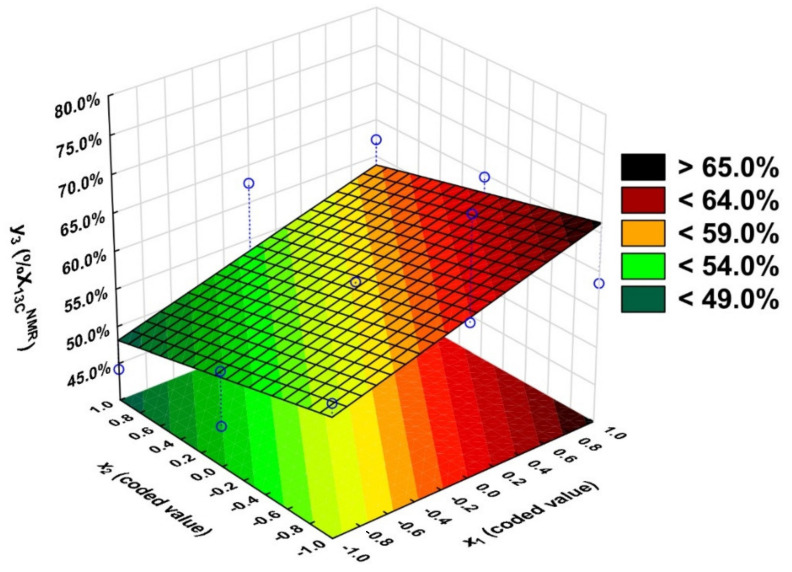
Dependence of itaconic anhydride conversion on the OH/COOH ratio (*x*_1_) and the temperature (*x*_2_), *x*_3_ = 1.

**Figure 13 molecules-27-04627-f013:**
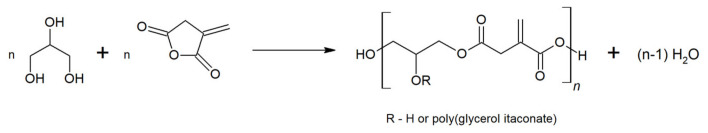
Synthesis of PGItc from glycerol and itaconic anhydride.

**Table 1 molecules-27-04627-t001:** Experimental matrix and results (Exp.—experimental, Calc.—calculated).

No.	Coded Variables			ED_titr_ [%]			ED_NMR_ [%]			%X_13C_^NMR^ [%]		
	*x* _1_	*x* _2_	*x* _3_	Exp.	Calc.	Rest	Exp.	Calc.	Rest	Exp.	Calc.	Rest
1	−1	−1	0	46.7	49.4	−2.7	68.1	66.7	1.4	57.4	55.6	1.8
2	1	−1	0	51.1	49.4	1.7	70.7	71.2	−0.5	58.5	66.3	−7.8
3	−1	1	0	48.4	49.4	−1.0	65.0	65.3	−0.3	44.2	48.1	−3.9
4	1	1	0	60.9	49.4	11.5	70.4	69.7	0.7	62.2	58.8	3.4
5	−1	0	−1	67.8	63.8	4.0	65.8	66.0	−0.2	52.7	51.8	0.9
6	1	0	−1	19.9	35.0	−15.1	69.7	70.4	−0.7	57.0	62.6	−5.6
7	−1	0	1	44.3	35.0	9.3	64.4	66.0	−1.6	45.4	51.9	−6.5
8	1	0	1	54.0	63.8	−9.8	70.2	70.4	−0.2	64.7	62.6	2.1
9	0	−1	−1	50.1	49.4	0.7	68.0	69.0	−1.0	60.3	61.0	−0.7
10	0	1	−1	48.9	49.4	−0.5	66.5	67.5	−1.0	51.0	53.5	−2.5
11	0	−1	1	43.5	49.4	−5.9	69.7	69.0	0.7	74.0	61.0	13.0
12	0	1	1	60.0	49.4	10.6	68.7	67.5	1.2	62.6	53.5	9.1
13	0	0	0	48.7	49.4	−0.7	68.5	68.2	0.3	57.3	57.2	0.1
14	0	0	0	47.9	49.4	−1.5	68.9	68.2	0.7	54.7	57.2	−2.5
15	0	0	0	48.5	49.4	−0.9	68.7	68.2	0.5	56.2	57.2	−1.0

**Table 2 molecules-27-04627-t002:** Calculated vs. experimental results.

Results	ED_tit_ [%]	ED_NMR_ [%]	%X_13C_^NMR^ [%]
Calculated	52.8	69.7	63.0
Experimental	62.3	70.4	62.8

**Table 3 molecules-27-04627-t003:** Box–Behnken Design—coding values.

Parameter	Natural Variable	Coded Value			Step
		−1	0	+1	
*x* _1_	OH/COOH ratio	0.5	1	1.5	0.5
*x* _2_	Temperature [°C]	110	125	140	15
*x* _3_	Time [h]	3	4	5	1

## Data Availability

Not applicable.

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
