# Peer review of "Optimisation of Glycerol and Itaconic Anhydride Polycondensation"

_molecules, 2022, doi:10.3390/molecules27144627_

Round 1
Reviewer 1 Report
Itaconate polyesters form rubber-like polymers upon crosslinking. They do not exhibit very high strength, but they can be used in tissue engineering as drug delivery hydrogels. The structure of the growing polymer can be controlled relatively simply by selecting the molar ratio of glycerol to acid or anhydride. The temperature and time of polycondensation influence the degree of esterification and branching, molecular weight, viscosity and mechanical properties of PGItc.
The work aims to optimise the synthesis of PGItc to obtain the highest possible degree of conversion. Optimisation experiments were performed with itaconic anhydride and glycerol using mathematical planning of experiments according to the Box-Behnken plan. The input variables of the process were the ratio of OH/COOH, temperature, and reaction time. The optimised output variables were: the degree of esterification (EDtitr), the degree of esterification calculated from the
analysis of 1H NMR spectra (EDNMR), and the degree of itaconic anhydride conversion, which was calculated based on 13C NMR spectra. Statistical models maximising the output variables were created. In each case, the significance of the changed synthesis parameters was checked. Optimal conditions for conducting the process were determined. The products with 62.3% and 70.4% esterification degrees were obtained. The degree of anhydride conversion was 62.8%.
The interpretation of the IR spectra is not convincing, but I think that the article can be published.
Author Response
Dear reviewer
Thank you for your review of the article we submitted. In this letter we would like to refer to your comments and suggestions.
1) “It is well known the tendency of itaconate to isomerize easily to the corresponding mesaconate derivative. Authors even have indicated in figure 8 the presence of mesaconate derivatives during the reaction. However, this isomerization has been not considered for the estimation of the conversion of itaconic anhydride. Furthermore. it could be also possible the formation of other species such as citraconic anhydride which can be formed by simple heating of itaconic anhydride. In addition, there is also the possibility of some additional side reactions that could happen, but they have not been considered either. For example, Ordelt saturation is a reaction hard to limit and authors have not provided any evidence about the absence of this transformation. The presence of all this series of undesired side reactions will affect the way the conversion is calculated. Therefore, authors should revise all the conversion values provided in the manuscript by considering the corresponding amount of the additional products produced.”
We are aware of the possible isomerization of itaconic fragments to mesaconic fragments and then to citraconic fragments, and we deliberately did not consider it in the calculations. The main purpose of the work was to determine the reactivity and total conversion of itaconic anhydride to derivatives containing double bond in side chain. The position of it is very significant for us in view of the planned Michael addition. We want to determine the effect of steric hindrance on the reactivity of the double bond, so only the double bonds in the side chain are the object of this research. The degree of isomerization is the object of our research and we plan to publish the results in the near future, in another article.
The mentioned Ordelt saturation is known to us, we calculated that it occurs in the 0.2-1.8% yield range. For the calculations, we used signals in the range of 2.5-3.1 ppm (Figure 1) and newly obtained signals in the unsaturated bond range. The calculated fraction of Ordelt saturation is the ratio of the resulting signals of protons after saturation, to the sum of the signals of protons after saturation and the signals of protons connected to the carbons in double bonds. We normalized the calculation by taking into account the number of protons generating a given signal.
Figure 1. 1 H NMR spectrum - Ordelt saturation range
2) “In equation inserted in line 145, on page 5, the value of the integral of the signal from the itaconic monoesters is multiplied by a half. What is the reason for that multiplier?”
There should not be a multiplier of 1/2 in the formula.
The error is due to copying a similar equation from our other work, in which 1/2 was the normalisation of the number of protons.
In this case, the normalisation is irrelevant. All the integrated signals refer to itaconic fragments, so each of them refers to 2 protons, which can be simplified.
3) “The discussion in section 2.2. about the statistical analysis performed should be more clearly presented. Authors have not indicated to the reader the aim of this analysis, which variables will be analysed and the reason for the choice of this particular statistical model employed. Some data provided later in the manuscript, but they should be mentioned earlier making the article easier to understand to the reader”
We have revised statistical analysis in Section 2.2, which is now beginning with an introduction why we chose the Box-Behnken plan and the advantages of mathematical methods for planning experiments. We also explained the choice of analysed variables.
4) “The introductory background provided in the abstract is quite extensive. It is almost half of the total content included. An abstract should be more simple, specific, and clearer summarizing the major points of the work and the conclusions drawn from the study conducted.”
We have also corrected the abstract. We have shortened the background, adding a few sentences about the research conducted.
5) “The interpretation of the IR spectra is not convincing, but I think that the article can be published” – second reviewer's comment
We have added brief descriptions of the characteristic vibrations to the image showing the FTIR spectrum of the product versus the substrate spectra. In this way, the FTIR spectrum has become more legible.
We hope we have answered all your suggestions. We have also corrected language and grammar.
We are sincerely looking forward to your feedback.
On behalf of the co-authors
DSc. PhD. Eng. Agnieszka Gadomska-Gajadhur
Faculty of Chemistry
Warsaw University of Technology
agnieszka.gajadhur@pw.edu.pl

Reviewer 2 Report
The present paper entitled “Optimisation of glycerol and itaconic anhydride polycondensation” by the authors Krzysztof Kolankowski, Magdalena MiÄ™tus, PaweÅ‚ RuÅ›kowski, and Agnieszka Gadomska-Gajadhur, describes the use of three mathematical models to evaluate the influence of various reactions parameters on the conversion and the degree of esterification in the reaction of itaconic anhydride and glycerol to produce poly(glycerol itaconate). For the optimization studies titration and NMR techniques were employed. The statistical model allows to determine the significance of each parameter studied and it also served to establish the optimal conditions for conducting the process.
After careful evaluation of the manuscript, I consider it will typically appeal to the reader of Molecules, and I would recommend it for publication in the journal if the following comments are considered:
· It is well known the tendency of itaconate to isomerize easily to the corresponding mesaconate derivative. Authors even have indicated in figure 8 the presence of mesaconate derivatives during the reaction. However, this isomerization has been not considered for the estimation of the conversion of itaconic anhydride. Furthermore. it could be also possible the formation of other species such as citraconic anhydride which can be formed by simple heating of itaconic anhydride. In addition, there is also the possibility of some additional side reactions that could happen, but they have not been considered either. For example, Ordelt saturation is a reaction hard to limit and authors have not provided any evidence about the absence of this transformation. The presence of all this series of undesired side reactions will affect the way the conversion is calculated. Therefore, authors should revise all the conversion values provided in the manuscript by considering the corresponding amount of the additional products produced.
· In equation inserted in line 145, on page 5, the value of the integral of the signal from the itaconic monoesters is multiplied by a half. What is the reason for that multiplier?
· The discussion in section 2.2. about the statistical analysis performed should be more clearly presented. Authors have not indicated to the reader the aim of this analysis, which variables will be analysed and the reason for the choice of this particular statistical model employed. Some data provided later in the manuscript, but they should be mentioned earlier making the article easier to understand to the reader.
· The introductory background provided in the abstract is quite extensive. It is almost half of the total content included. An abstract should be more simple, specific, and clearer summarizing the major points of the work and the conclusions drawn from the study conducted.
Author Response
.

Round 2
Reviewer 2 Report
Authors have made the appropriate corrections, and their explanations concerning the issues addressed in my comments seems reasonable to me.
There is only one point that I think is not still clear along the manuscript. Authors have explained in the conclusion section the presence of Ordelt side reactions but there is no reference along the entire article to that transformation. They should clearly explain that issue and to include in the SI how they have estimated the ratio of that undesirable reaction.
Author Response
Dear reviewer
We have slightly expanded the fragment about Ordelt saturation and moved it to the section with spectral analysis. We mentioned the possible isomerization of itaconic fragments to mesaconic fragments. For both side reactions, we calculated their yields. We have included the 1H NMR spectrum and the calculation methodology in the supplementary information.
We have also improved the language aspect.
We sincerely hope that you will be satisfied
Kind regards
On behalf of the co-authors
DSc. PhD. Eng. Agnieszka Gadomska-Gajadhur
Faculty of Chemistry
Warsaw University of Technology
agnieszka.gajadhur@pw.edu.pl